# Differences in Results and Related Factors between Hospital-At-Home Modalities in Catalonia: A Cross-Sectional Study

**DOI:** 10.3390/jcm9051461

**Published:** 2020-05-13

**Authors:** Jorge Arias-de la Torre, Evangelia Anna María Zioga, Lizza Macorigh, Laura Muñoz, Oriol Estrada, Montse Mias, Maria-Dolors Estrada, Elisa Puigdomenech, Jose M. Valderas, Vicente Martín, Antonio J. Molina, Mireia Espallargues

**Affiliations:** 1Agència de Qualitat i Avaluació Sanitàries de Catalunya (AQuAS), 08005 Barcelona, Spain; lmunyoz@gencat.cat (L.M.); mmias@gencat.cat (M.M.); destrada@gencat.cat (M.-D.E.); epuigdomenech@gencat.cat (E.P.); mespallargues@gencat.cat (M.E.); 2CIBER Epidemiología y Salud Pública (CIBERESP), 28029 Madrid, Spain; vicente.martin@unileon.es; 3Instituto de Biomedicina (IBIOMED). Universidad de León, 24004 León, Spain; ajmolt@unileon.es; 4Departamento de Medicina Interna, Hospital Dos de Mayo, 08025 Barcelona, Spain; eamzioga@gmail.com; 5Departamento de Medicina Interna, Hospital de Granollers, 08402 Barcelona, Spain; lizzamacorigh@gmail.com; 6Red de Investigación en Servicios de Salud en Enfermedades Crónicas (REDISSEC), 28029 Madrid, Spain; 7Dirección de Procesos Asistenciales y Alianzas. Gerencia Territorial Metropolitana Nord, Institut Català de la Salut, 08007 Barcelona, Spain; oestrada@gencat.cat; 8Health Services and Policy Research Group, University of Exeter Medical School, Exeter EX2 4TE, UK; j.m.valderas@exeter.ac.uk

**Keywords:** hospital-at-home, readmission, mortality, average stay, cross-sectional study

## Abstract

Hospital-at-home (HaH) is a healthcare modality that provides active treatment by healthcare staff in the patient’s home for a condition that would otherwise require hospitalization. The aims were to describe the characteristics of different types of hospital-at-home (HaH), assess their results, and examine which factors could be related to these results. A cross-sectional study based on data from all 2014 HaH contacts from Catalonia was designed. The following HaH modalities were considered—admission avoidance (*n* = 7214; 75.1%) and early assisted discharge (*n* = 2387; 24.9%). The main outcome indicators were readmission, mortality, and length of stay (days). Multivariable models were fitted to assess the association between explanatory factors and outcomes. Hospital admission avoidance is a scheme in which, instead of being admitted to acute care hospitals, patients are directly treated in their own homes. Early assisted discharge is a scheme in which hospital in-care patients continue their treatment at home. In the hospital avoidance modality, there were 8.3% readmissions, 0.9% mortality, and a mean length of stay (SD) of 9.6 (10.6) days. In the early assisted discharge modality, these figures were 7.9%, 0.5%, and 9.8 (11.1), respectively. In both modalities, readmission and mean length of stay were related to comorbidity and type of hospital, and mortality with age. The results of HaH in Catalonia are similar to those observed in other contexts. The factors related to these results identified might help to improve the effectiveness and efficiency of the different HaH modalities.

## 1. Introduction

Hospital-at-home (HaH) is a healthcare modality that, for a limited period of time, provides active treatment by healthcare staff in the patient’s home for a condition that would otherwise require hospitalization [1,2,3,4]. Unlike other home care services that are focused on less severe patients and are carried out by primary care staff, HaH is specialized (secondary) care and, therefore, is performed by different hospital specialists. Previous studies have described two basic types of HaH: hospital admission avoidance and early assisted discharge [5,6,7]. The admission avoidance model is usually employed with elderly individuals who, instead of being admitted to acute care hospitals, are treated at home [6,8,9]. The model mainly focuses on short-term interventions (days) for the acute phase of an illness. With respect to admission to this HaH modality, patients are mostly included after being attended by emergency services or, less commonly, after a referral from their family doctor. In contrast, the early assisted discharge HaH model is for hospitalized patients who are able to continue their treatment at home, thus reducing the duration of their stay [7]. 

Within the context of Catalonia, since 1985, the HaH model has been officially recognized as a healthcare activity or service. In spite of this legal framework, the posterior evolution of the healthcare system within the territory has not led to the program’s homogenous development [8,10,11,12,13,14]. This has resulted in the appearance of HaH units without any pre-established or defined resource structures and with varying service portfolios, all of which have hindered a common evaluation. Nevertheless, in spite of the lack of homogeneity, the patients included in HaH programs can be categorized according to two modalities: hospital admission avoidance and early assisted discharge. Although such a classification is very general [5,6,7,14], it does permit an evaluation of their respective results at a population level and the determination of factors potentially related to each of the modalities. The delimitation of the factors associated with such results for each modality in a particular context might allow populations that could have a higher potential benefit of receiving this attention to be defined. This identification could be the first step for causal research to define and establish the most effective and efficient healthcare circuits, thus helping to improve the modalities’ results and avoiding unnecessary costs to the healthcare systems.

The objectives of this study were: (1) to describe the contact characteristics of both HaH modalities (admission avoidance and early assisted discharge) in Catalonia during 2014; (2) to evaluate the rates of readmission, mortality, and mean length of stay for each of the modalities; and (3) to examine which factors could be related to their results. 

## 2. Experimental Section

### 2.1. Study Design and Population

An exploratory cross-sectional study based on the Minimum Basic Data Set from Acute-care Hospitals (MBDSHA) was performed. The MBDSHA included 24 public hospitals in the Catalonian territory and HaH contacts for 2014 [15]. A contact was every time a patient received any kind of treatment from commencement to finalization. The same individual could present more than one contact during the study period. Programmed contacts with a specific diagnosis according to the International Classification of Diseases, Ninth Revision, Clinical Modification (ICD-9-CM) were included (*n* = 9805) [16]. Those who belonged to a diagnostic group with an insufficient number of contacts for robustness (*n* = 95, 1.0%), lacked an identification number (*n* = 49, 0.5%), had no established age (*n* = 2, <0.1%), and whose dates of admission and discharge were wrongly codified (58, 0.6%), were excluded. Finally, a total sample of 9601 HaH contacts was considered for analysis—7214 (75.1%) admission avoidance and 2387 (24.9%) early assisted discharge. 

### 2.2. Main Outcomes

Three indicators were established as the main outcomes for the exploratory analysis. These indicators were selected both based on their relevance to assessing the performance of the HaH modalities, as well as based on the recommendations for the assessment of HaH from previous systematic reviews [6,7]. 

Readmission: based on the definition of the Centers for Medicare and Medicaid Services (CMS) [17], this indicator was defined as a consecutive HaH contact, either HaH or conventional hospitalization (CH), related to the first HaH attention in a period ≤ 30 days starting from 1 January 2014 up to 30 January 2015 (to consider those contacts from December 2014).

Mortality prior to discharge: HaH contacts in which the patient status at discharge was death.

Mean length of stay: for the admission avoidance modality, duration (days) of HaH contact from the date of program admission to finalization. For early assisted discharge, the duration was a combination of the immediately preceding contact of CH and the HaH one, taking it from the CH contact date of admission to HaH finalization.

### 2.3. Contact Characteristics 

Sex: male and female.Age (years): considered a continuous variable.Diagnosis: categorized according to the ICD-9-CM chapters [16].Comorbidity according to the Charlson Comorbidity Index (CCI) [18,19]. The CCI, considered to be an objective measurement of an individual’s general state of health, is employed to predict mortality in terms of the patient’s comorbidity. General comorbidity is calculated through the weight assigned to the presence of each of the 19 conditions making up the index. The results are classified as 0 or 1, 2, and ≥3.Type of hospital (according to the portfolio of services offered in the hospital itself, irrespective of the patient’s territorial assignment): reference hospital, district hospital, general high-technology hospital, and high-resolution hospital.Number of contacts per patient (number of HaH episodes): 1 or more than 1.

### 2.4. Data Analysis

A descriptive analysis of the characteristics of the contacts according to the HaH modality was performed and the results were evaluated at the bivariable level. To compare the possible differences in the explicative variables between the two modalities, Chi-square and Fisher’s exact tests were employed for the categorical variables, and the Mann–Whitney U test for age and mean length of stay due to the lack of normality of their distributions. The selected outcomes were then calculated for each modality and the association between the contacts’ characteristics, and each of these indicators was assessed with multivariable models. Due to the characteristics of the variables considered as outcomes, logistic regression models were fitted for readmission and mortality, and Poisson models for mean length of stay. From these results, the β coefficients and their respective 95% confidence intervals (95% CI) were obtained, and their exponential was presented to aid interpretation (the Odds Ratio in logistic regression models and ratios for Poisson models). All multivariable models were done individually for the two HaH modalities and adjusted for sex, age, comorbidity (CCI), and the type of hospital. The absence of relevant interactions between explanatory variables was tested using a Chunk test. All analyses were carried out with the STATA v.14^®^ (StataCorp, College Station, TX, US) [20] software and statistical significance was set to α = 0.05. 

## 3. Results

Table 1 shows the contacts’ characteristics in terms of their HaH modality. The contact frequency for admission avoidance during 2014 in Catalonia (*n* = 7214) was greater than that of early assisted discharge (*n* = 2387). Differences were observed between the two HaH modalities for sex, diagnostic group, and type of hospital. The diagnostic groups with the most contacts were diseases of the respiratory system. With respect to the indicators calculated for each of the HaH modalities (Table 2), while significant differences were not found for readmissions or for mean length of stay, differences (*p* = 0.04) in mortality before discharge were found.

Table 3 shows that in the admission avoidance, readmission was related to a CCI ≥3 (exp(β): 1.69; CI95%: 1.39–2.07), type of hospital, and age (exp(β): 1.02; CI95%: 1.01–1.02); mortality prior to discharge was related to a CCI ≥3 (exp(β): 1.89; CI95%: 1.08–3.31), to be treated in a high-resolution hospital (exp (β): 2.11; CI95%: 1.04–4.27), and age (exp (β): 1.07; CI95%: 1.05–1.10). The mean length of stay was greater in women than in men (exp(β): 0.94; CI95%: 0.92–0.96), and was related to the CCI, type of hospital, and age. For the early assisted discharge modality, readmission was related to the CCI and type of hospital, mortality was associated with age (exp (β): 1.11; CI95%: 1.04–1.20), and the mean length of stay was related to sex, having a CCI ≥3 (exp (β): 1.07; CI95%: 1.03–1.10), type of hospital, and age.

## 4. Discussion

Our findings show that in 2014, in Catalonia, the results for readmission, mortality, and mean length of stay for the two HaH modalities, in spite of their heterogeneous development, could be similar to those observed in previous studies throughout the world [5,6,7,21]. In addition, irrespective of the modality, it was observed that comorbidity and the type of hospital are related to readmission and mean length of stay whilst the patient’s age is linked to mortality. Such information could serve as a starting point to guide further research on causal relationships between the studied variables and the results of HaH. This research could help more precisely define the most suitable healthcare circuits, and the type of patients who could most benefit from the different HaH modalities. 

Previous studies comparing HaH and CH [5,13,22,23] have reported that home care can have similar results to CH and would save both human and economic resources [24,25,26,27]. In this sense, both the results obtained and the available evidence suggest that, provided the patient’s indication permit it and new causal research confirm them, HaH treatment could be a suitable, effective, and perhaps efficient alternative to CH [10,21,25,28,29]. Furthermore, it should be noted that mental health contacts were not included. Mental health HaH units have a strong psychosocial component involving further than physicians and nurses, clinical psychologists, and social workers. Besides, HaH for mental health usually is organized in a different way than for other conditions [5,30]. These differences make difficult, and possibly biased, the comparison of outcomes between mental health HaH services with HaH services for physical conditions. Despite these differences, new evidence focused on mental health, could be valuable in assessing the results of HaH from a wider perspective.

Regarding the HaH modalities, we observed that whilst admission avoidance was more frequent in less complex hospitals, the early discharge was more common in more complex ones. Despite this, it is important to note that admission avoidance was applied more often than early discharge in both types of hospitals. More complex or more serious cases could be addressed to a greater extent in high-technology and high-resolution hospitals. These hospitals generally have a wide range of technology and services, which results in a greater demand for healthcare. Thus, the results obtained could allow the hypothesis that adopting an early assisted discharge program in hospitals of greater complexity, providing the patient’s indication allows it, could result in the optimization of services that are only available in these institutions. Besides, we can hypothesize that the use of admission avoidance or early discharge schemes could have different outcomes for the treatment of specific conditions [7,9,11]. Further causal studies centered on the early assisted discharge modality in high-technology and high-resolution hospitals accounting for different specific diseases could help assess the optimization of such specialized services.

With respect to the factors related to the considered outcomes, in both modalities, it was observed that whilst readmission and mean length of stay were associated with the CCI and type of hospital, mortality was related to the patient’s age. In addition, in the admission avoidance modality, mortality was only related to comorbidity in patients with the highest CCI scores. Nevertheless, in spite of being non-significant, the relationship in patients with a lower CCI was expected as found, possibly predicting risk. In this manner, as reported by previous studies [6,31], our findings suggest that comorbidity could be a particularly relevant factor when choosing this modality. Further longitudinal studies aimed to compare both modalities with a greater sample size could be valuable in confirming this hypothesis, and thus, better aid in the indication and improve the results. Moreover, readmissions were considered globally (including subsequent contacts both of HaH and CH), and therefore, new research accounting for the type of readmission could be relevant for the evaluation of HaH services. 

As limitations of the study, it is worth highlighting the limitation related to the use of data from 2014. Despite this, since this year there are no significant changes in the characteristics of the study population [15], we deem it reasonable to assume that the results could also be relevant currently. Another limitation is the possible reductionism in which the classification of HaH modalities falls. Nevertheless, this classification has been previously employed on numerous occasions when evaluating HaH [6,7,23]. In addition, given the heterogeneity observed amongst the HaH units in Catalonia, it permits a general evaluation of the factors potentially related to their results at a population level. Another limitation is related to the cross-sectional design of the study. It does not allow the direction of the relationship amongst the variables to be established, precluding the establishment of causal relationships between variables. Nevertheless, as our study is an exploratory analysis to find out what factors could be related to the proposed outcomes, our results could serve to establish hypotheses on causal relationships and could also be a starting point for future causal research. Furthermore, taking into account the hospitals included, and their type, a multi-level structure of the data is suggested. Despite this possible multi-level structure, our study could serve to determine the factors potentially related to the outcomes selected globally (for the HaH modalities as a whole) and as a starting point to further multi-level analyses. About the generalizability of the results, our study is focused only in Catalonia, and therefore, the capability to extrapolate the results to other contexts might be compromised. Nevertheless, as the HaH schemes are similar to those from other regions of Spain and from other countries [6,7], the results could be, to a certain extent, extrapolated to these contexts. Lastly, the limitation regarding the variables included should be mentioned. The inclusion of other variables not considered related to the patient or hospital, like social support, length of contacts, or specific treatments, could help more accurately define what factors could be associated with the indicators that were analyzed. However, we consider that the included variables allow for the adjustment of more parsimonious and easily interpretable models. In addition, these variables cover basic aspects of both the patients and the care process, to a large extent, and an analysis of these factors is also a basic previous step in conducting studies in greater depth.

## 5. Conclusions

In spite of the heterogeneity of HaH development in Catalonia, our findings in terms of readmission, mortality, and mean length of stay are in line with previous studies in other settings [5,6,7,10]. In addition, the results provide new evidence about which factors might be related to these results for admission avoidance and early assisted discharge HaH modalities. Further causal research based on the results obtained and taking other more specific HaH indicators into account, like cost-saving with respect to the CH or freeing-up beds, could help define more precisely the type of patient and care circuit, thus increasing the effectivity and efficiency of the different modalities. Therefore, this study shows that HaH could be an effective and efficient healthcare modality to provide treatments for patients eligible for it and, possibly, reducing costs for the healthcare systems.

## Figures and Tables

**Table 1 jcm-09-01461-t001:** Characteristics of the contacts, hospital, and care process according to the hospital-at-home modality.

	Admission Avoidance (*n* = 7214)	Early Discharge (*n* = 2387)	*p Value*
	*n* (%)	*n* (%)	
**Mean age in years (SD)**	69.9 (17.2)	69.6 (16.8)	0.309
**Sex**			0.013
Male	3800 (52.7)	1327 (55.6)	
Female	3414 (47.3)	1060 (44.4)	
**Diagnostic group**			<0.001
Respiratory system diseases	2440 (33.8)	780 (32.7)	
Genitourinary system diseases	976 (13.5)	412 (17.3)	
Circulatory system diseases	1030 (14.3)	234 (9.8)	
Osteomioarticular and connective tissue system diseases	747 (10.4)	137 (5.7)	
Lesions and intoxications	499 (6.9)	250 (10.5)	
Digestive tract diseases	470 (6.5)	268 (11.2)	
Skin and subcutaneous tissue diseases	363 (5.0)	79 (3.3)	
Neoplasms	247 (3.4)	53 (2.2)	
Central nervous system and sense organ diseases	131 (1.8)	62 (2.6)	
Badly defined symptoms, signs, and conditions	138 (1.9)	46 (1.9)	
Endocrine, nutritional and metabolic diseases, and immunity disorders	106 (1.5)	42 (1.8)	
Blood and hematopoietic organ diseases	67 (0.9)	24 (1.0)	
**Comorbidity (Charlson Index)**			0.626
0 or 1	4479 (62.1)	1477 (61.9)	
2	1168 (16.2)	405 (17.0)	
≥ 3	1567 (21.7)	505 (21.2)	
**Type of hospital**			<0.001
Reference hospital	2673 (37.1)	534 (22.4)	
District hospital	2322 (32.2)	738 (30.9)	
High-technology general hospital	1029 (14.3)	608 (25.5)	
High-resolution reference hospital	1190 (16.5)	507 (21.2)	
**Number of contacts per patient**			0.254
1	5383 (74.6)	1809 (75.8)	
>1	1831 (25.4)	578 (24.2)	

*n*: number of contacts; %: percentage of contacts; SD: standard deviation. *p*: *p*-value obtained with Chi-square test and Fisher’s test for categorical variables, and Mann–Whitney *U* for age.

**Table 2 jcm-09-01461-t002:** Readmission, mortality, and mean length of stay according to the hospital-at-home modality.

	Admission Avoidance (*n* = 7214)	Early Discharge (*n* = 2387)	*p*-Value
	*n* (%)	*n* (%)	
**Readmission**			0.524
No	6613 (91.7)	2198 (92.1)	
Yes	601 (8.3)	189 (7.9)	
**Mortality prior to discharge**			0.040
No exitus	7146 (99.1)	2375 (99.5)	
Exitus	68 (0.9)	12 (0.5)	
**Mean contact stay (SD) in days**	9.6 (10.6)	9.8 (11.1)	0.059
**Total days of stay**	68,934	23,460	

*n*: number of contacts; %: percentage of contacts; SD: standard deviation. *p*: *p*-value obtained with the Chi-square test and Fisher’s test for categorical variables, and Mann–Whitney *U* for mean length of stay.

**Table 3 jcm-09-01461-t003:** Factors related to readmission, mortality, and mean length of stay according to the hospital-at-home modality.

	Readmission				Mortality Prior to Discharge				Mean Length of Stay			
	*n*	%	aOR (CI95%)	*p*	*n*	%	aOR (CI95%)	*p*	mean	SD	Ratio (CI95%)	*p*
***Admission avoidance***												
Sex												
Male	302	8.0	1.00		37	1.0	1.00		9.9	11.3	1.00	
Female	299	8.8	1.17 (0.99–1.39)	0.065	31	0.9	0.87 (0.53–1.44)	0.582	9.1	9.8	0.94 (0.92–0.96)	<0.001
**Charlson Index**												
0 or 1	304	6.8	1.00		28	0.6	1.00		8.8	10.0	1.00	
2	108	9.3	1.22 (0.97–1.55)	0.083	16	1.4	1.67 (0.90–3.11)	0.106	10.6	12.3	1.20 (1.18–1.22)	<0.001
≥ 3	189	12.1	1.69 (1.39–2.07)	<0.001	24	1.5	1.89 (1.08–3.31)	0.027	10.8	10.6	1.26 (1.24–1.28)	<0.001
**Type of hospital**												
Reference hospital	156	5.8	1.00		15	0.6	1.00		9.6	10.4	1.00	
District hospital	217	9.4	1.48 (1.19–1.83)	<0.001	28	1.2	1.54 (0.81–2.91)	0.182	7.1	5.7	0.74 (0.72–0.75)	<0.001
High-technology general hospital	106	10.3	1.62 (1.25–2.11)	<0.001	8	0.8	1.01 (0.43–2.40)	0.975	11.7	11.4	1.19 (1.17–1.22)	<0.001
High-resolution hospital	122	10.3	1.76 (1.37–2.27)	<0.001	17	1.4	2.11 (1.04–4.27)	0.037	12.4	15.5	1.30 (1.27–1.33)	<0.001
**Age**			1.02 (1.01–1.02)	<0.001			1.07 (1.05–1.10)	<0.001			1.00 (1.00–1.00)	<0.001
***Early discharge***												
**Sex**												
Male	117	8.8	1.00		5	0.4	1.00		9.8	8.7	1.00	
Female	72	6.8	0.82 (0.60–1.12)	0.212	7	0.7	1.23 (0.34–4.04)	0.729	9.9	13.5	1.06 (1.04–1.09)	<0.001
**Charlson Index**												
0 or 1	93	6.3	1.00		8	0.5	1.00		9.7	10.8	1.00	
2	46	11.4	1.70 (1.16–2.45)	0.006	3	0.7	1.04 (0.27–3.99)	0.954	9.8	12.3	1.00 (0.97–1.05)	0.596
≥ 3	50	9.9	1.52 (1.05–2.21)	0.028	1	0.2	0.28 (0.03–2.29)	0.235	10.2	11.1	1.07 (1.03–1.10)	<0.001
**Type of hospital**												
Reference hospital	18	3.4	1.00		2	0.4	1.00		9.4	9.1	1.00	
District hospital	52	7.1	2.24 (1.29–3.88)	0.004	4	0.5	1.30 (0.24–7.21)	0.761	8.7	13.0	0.92 (0.89–0.95)	<0.001
High-technology general hospital	68	11.2	3.60 (2.10–6.16)	<0.001	2	0.3	1.27 (0.17–9.21)	0.814	13.0	12.1	1.37 (1.32–1.42)	<0.001
High-resolution hospital	51	10.1	3.20 (1.84–5.56)	<0.001	4	0.8	2.24 (0.40–12.40)	0.356	8.1	7.4	0.86 (0.82–0.89)	<0.001
**Age**			1.01 (1.00–1.02)	0.204			1.11 (1.04–1.20)	0.002			1.00 (1.00–1.00)	0.002

*n*: number of contacts with a positive indicator; % percentage of contacts with a positive indicator; mean: mean of the duration of days of contact. SD: standard deviation. aOR/Ratio (CI95%): adjusted Odds Ratio/adjusted Ratio and 95% confidence Interval adjusted by the hospital-at-home modality, sex, age, comorbidity (Charlson index), and type of hospital; *p*: *p*-value based on the multivariate model.

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
