# Peer review of "Differences in Results and Related Factors between Hospital-At-Home Modalities in Catalonia: A Cross-Sectional Study"

_jcm, 2020, doi:10.3390/jcm9051461_

Round 1

Reviewer 1 Report

Differences in results and related factors between Hospital-at-Home modalities in Catalonia: a cross-sectional study

This well-structure and well-written manuscript describes the use of Hospital-at-home (HaH) services throughout Catalonia for both HaH facilities relating to admission avoidance and HaH facilities relating to early assisted discharge. Cross-sectional associations between potential explanatory factors are also described. The results of this paper suggest that HaH services hold promise and therefore the results of this paper are clinically useful. Some further qork is required, however, before this manuscript is ready for publication.

Abstract

If possible, could a brief definition of the two modalities (admission avoidance and hospital discharge) be provided in the Abstract?

Experimental section

Could some clarity be provided as to why £30 day (i.e. early) readmission was selected as an outcome? Is there a reason as to why all readmissions within 2014 related to the first contact also were not assessed?

In terms of readmissions related to use of HaH for admission avoidance, would it be possible to have two readmission outcomes – one relating to readmittance to HaH service and one relating to conventional hospital admissions? This way, one could quantify how successful HaH for admission avoidance actually is for avoiding conventional hospital admissions.

Mental health was not included as a diagnostic group. I assume this is because HaH is not utilised for mental health conditions? Is early assisted discharge used for psychiatric conditions? It would be good to clarify this for a reader who is not familiar with this healthcare modality (perhaps in the Discussion).

Table 1 and Table 2 – make sure to be consistent with decimal places (see ‘Badly defined symptoms, signs, and conditions’ in Table 1) and avoid using commas as decimal places (see Table 2).

Discussion

In the Discussion, the authors say that HaH treatment might be a suitable alternative to conventional hospitalisation. It would be clinically useful and incredibly interesting to see if HaH might be more successful/useful for some diagnostic groups more than others, perhaps included as supplementary analysis?

Are there any limitations relating to the generalisability of results?

Author Response

# Reviewer 1

This well-structure and well-written manuscript describes the use of Hospital-at-home (HaH) services throughout Catalonia for both HaH facilities relating to admission avoidance and HaH facilities relating to early assisted discharge. Cross-sectional associations between potential explanatory factors are also described. The results of this paper suggest that HaH services hold promise and therefore the results of this paper are clinically useful. Some further qork is required, however, before this manuscript is ready for publication.

R: We want to thank you for the time spent reading our manuscript and for your valuable comments. Below you can find a point by point response detailing the changes made. We hope these revisions meet your expectations. 

Abstract

If possible, could a brief definition of the two modalities (admission avoidance and hospital discharge) be provided in the Abstract?

R: Thank you for this comment. We have now included a brief definition of the two modalities in the abstract.

  • Changes in text: Please see lines 30 to 32 of the abstract

Experimental section

Could some clarity be provided as to why £30 day (i.e. early) readmission was selected as an outcome? Is there a reason as to why all readmissions within 2014 related to the first contact also were not assessed?

R: Thank you for this suggestion. We have now included the reason for the selection of the 30-day value. This value was selected based on a definition commonly used in health services research and planification of health resources. This is the definition proposed by the Centers for Medicare and Medicaid Services (CMS) of the U.S., and thus we have now included also a reference to support its use.

About the second question, please accept our apologies because the definition of readmission provided was wordy and could lead to misunderstandings. We have now reworded it to clarify that all readmissions within 2014 related to the first contact were considered in the analyses.

  • Changes in text: Please see lines 93 to 97 of the experimental section
  • Please see reference 17

In terms of readmissions related to use of HaH for admission avoidance, would it be possible to have two readmission outcomes – one relating to readmittance to HaH service and one relating to conventional hospital admissions? This way, one could quantify how successful HaH for admission avoidance actually is for avoiding conventional hospital admissions.

R: We thank the reviewer for this appreciation. To perform this analysis, it might be convenient take into account different factors not contemplated in our study such as the severity of specific illnesses or, at least, separate specific acute and chronic conditions within the ICD groups. In spite of these difficulties in the analysis and our limitation in this aspect, as we completely agree that separate by the conventional and HaH readmissions could be a very relevant next step for the evaluation of HaH services, we have now included this suggestion within the discussion.

  •  Changes in text: Please see lines 240 to 242 of the discussion.

Mental health was not included as a diagnostic group. I assume this is because HaH is not utilised for mental health conditions? Is early assisted discharge used for psychiatric conditions? It would be good to clarify this for a reader who is not familiar with this healthcare modality (perhaps in the Discussion).

R: Thank you for this comment. There is a reason to not include mental disorders in this study and is that Hospital at Home services for mental health are a separate entity and completely different that for physical conditions. We have to explain that the HaH circuit for mental health is organized in a different way in Catalonia (and in the rest of Spain) than the circuit for other conditions and, as they are not comparable, their evaluation should be done separately than for other medical conditions. HaH units in mental health have a strong psychosocial component involving further than physicians and nurses, clinical psychologists and social workers making difficult to compare outcomes with the HaH services for physical conditions. These aspects were now included as part of the discussion.

  • Changes in text: Please see lines 212 to 218 of the discussion

Table 1 and Table 2 – make sure to be consistent with decimal places (see ‘Badly defined symptoms, signs, and conditions’ in Table 1) and avoid using commas as decimal places (see Table 2).

R: We apologize for these mistakes. Tables are now correct and consistent.

  • Changes: Please see Table 1 and Table 2

Discussion

In the Discussion, the authors say that HaH treatment might be a suitable alternative to conventional hospitalisation. It would be clinically useful and incredibly interesting to see if HaH might be more successful/useful for some diagnostic groups more than others, perhaps included as supplementary analysis?

R: Thank you for this appreciation. Due to the low number of individuals in some of the diagnostic groups (particularly on mortality), we are not able to do these supplementary analyses. After carefully think about and due to the lack of robustness of these analyses, we have serious concerns about the validity of their results and about present them.  However, we have now included some discussion about the usefulness of new research focused on differences in result of HaH according to the diagnostic group.

  • Changes in text: Please see lines 227 to 230 of the discussion.

Are there any limitations relating to the generalisability of results?

R: Thank you for pointing out this limitation. We have now included it within the discussion.

  • Changes in text: Please see lines 258 to 262 of the discussion.

Reviewer 2 Report

It is good to see study reports on a new form of home care, hospital at home. The findings are of interest and relevance in most countries. There are a few limitations or additional considerations that should be outlined or discussed in the paper:

  1. what was actually done to the patients in their homes? Who provided the care in the home? How long was each visit?
  2. how does Hospital at Home differ from home care services in Spain?
  3. a major limitation is that 2014 data were used, that should be stated as health care and health system and population changes could have occurred since then, and also changes in the Hospital at Home model. Please tell us of these changes, or if there are non - please state that.
  4. in the end, a statement needs to be made about the value of Hospital at Home care -  is it an important new service, or an add on costly service?

Author Response

# Reviewer 2

It is good to see study reports on a new form of home care, hospital at home. The findings are of interest and relevance in most countries. There are a few limitations or additional considerations that should be outlined or discussed in the paper:

R: Thank you for the positive evaluation of our research. We would also like to acknowledge the time spent on revising our manuscript and the valuable comments given. We have made the changes proposed and have included a brief explanation for each.

  1. What was actually done to the patients in their homes? Who provided the care in the home? How long was each visit?
  2. how does Hospital at Home differ from home care services in Spain?

R: We thank the reviewer for these appreciations. As they are closely related, please let us answer all in the same point.

There are two differences particularly relevant between HaH and other healthcare services provided in the patients’ home in Spain: the level of specialisation and the type of patients attended. About the first aspect, HaH is specialised (secondary) care and thus should be performed by different hospital specialists. In difference, other home care services were carried out by primary care staff. The second aspect is that, in difference with other home care services that usually attend less severe illnesses or conditions, HaH patients usually have conditions that would otherwise require a hospital admission.

The duration of the visits, both of HaH and other types of care is variable and depends  on different factors such as the workload of the specialized (the number of patients per day assigned), the severity of the condition to be treated and the type of treatment. Consequently, we consider reasonable to think that determine and compare the length of HaH visits or make a comparison of this aspect between HaH modalities, is beyond the scope of this article and could be an interesting topic for further research.

These explanations are now included in the introduction and discussion.

  • Changes in text: Please see lines 30 to 32 of the abstract
  • Changes in text: Please see lines 47 to 49 of the introduction.
  • Changes in text: Please see line 264 of the discussion.

  1. A major limitation is that 2014 data were used, that should be stated as health care and health system and population changes could have occurred since then, and also changes in the Hospital at Home model. Please tell us of these changes, or if there are non - please state that.

R: Thank you for this comment. We have now included this limitation and an explanation of it within the discussion.

  • Changes in text: Please see lines 243 to 246 of the discussion.

  1. In the end, a statement needs to be made about the value of Hospital at Home care - is it an important new service, or an add on costly service?

R: Thank you for this suggestion. We have now finished the main text highlighting the relevance of HaH as it could be an effective an efficient healthcare modality to provide treatments for patients eligible for it and, possibly, reducing costs for the healthcare systems.

  • Changes in text: Please see lines 278 to 279 of the discussion.
